# Pharmacologic Control of CAR T Cells

**DOI:** 10.3390/ijms22094320

**Published:** 2021-04-21

**Authors:** Benjamin Caulier, Jorrit M. Enserink, Sébastien Wälchli

**Affiliations:** 1Translational Research Unit, Section for Cellular Therapy, Department of Oncology, Oslo University Hospital, 0379 Oslo, Norway; b.w.caulier@medisin.uio.no; 2Center for Cancer Cell Reprogramming (CanCell), Institute for Clinical Medicine, Faculty of Medicine, University of Oslo, 0379 Oslo, Norway; jorrit.enserink@ibv.uio.no; 3Department of Molecular Cell Biology, Institute for Cancer Research, The Norwegian Radium Hospital, Montebello, 0379 Oslo, Norway; 4Section for Biochemistry and Molecular Biology, Faculty of Mathematics and Natural Sciences, University of Oslo, 0379 Oslo, Norway

**Keywords:** chimeric antigen receptor, small molecules, drugs, kinase inhibitors, CAR T cell, immunotherapy

## Abstract

Chimeric antigen receptor (CAR) therapy is a promising modality for the treatment of advanced cancers that are otherwise incurable. During the last decade, different centers worldwide have tested the anti-CD19 CAR T cells and shown clinical benefits in the treatment of B cell tumors. However, despite these encouraging results, CAR treatment has also been found to lead to serious side effects and capricious response profiles in patients. In addition, the CD19 CAR success has been difficult to reproduce for other types of malignancy. The appearance of resistant tumor variants, the lack of antigen specificity, and the occurrence of severe adverse effects due to over-stimulation of the therapeutic cells have been identified as the major impediments. This has motivated a growing interest in developing strategies to overcome these hurdles through CAR control. Among them, the combination of small molecules and approved drugs with CAR T cells has been investigated. These have been exploited to induce a synergistic anti-cancer effect but also to control the presence of the CAR T cells or tune the therapeutic activity. In the present review, we discuss opportunistic and rational approaches involving drugs featuring anti-cancer efficacy and CAR-adjustable effect.

## 1. Introduction

Chimeric antigen receptor (CAR)-modified T cells therapy has emerged almost two decades ago as an innovative cancer immunotherapy and was approved four years ago to treat certain B cell malignancies [1]. CAR therapy is part of a larger treatment family known as adoptive cell transfer (ACT). Here, T cells are genetically modified to express a receptor that replaces the function of the endogenous T cell receptor (TCR) to specifically bind a defined tumor antigen. CAR is a synthetic molecule restricted to surface antigen recognition, which upon binding will cluster and stimulate immune cellular functions. Although different CAR generations were developed to improve effector function and persistence, receptors are all composed of an extracellular recognition domain, a transmembrane anchoring domain, and an intracellular signaling domain [2,3]. The extracellular module is usually made of an antibody’s single-chain variable fragment (scFv) followed by a hinge region. The transmembrane domain ensures CAR distribution at the plasma membrane and is connected to the intracellular domain, which is composed of a combination of signaling domains of the TCR machinery [1]. Thus, redirected CAR T cells are activated upon antigen challenge to specifically kill the tumor by cytokine release and cytotoxicity. Therefore, we can distinguish two major factors that will influence the success of the therapy: the quality of the targeted antigen and the efficacy of the intracellular stimulation.

One of the first CAR targets was CD19, a pan B-lymphocyte marker, for which tremendous responses were obtained in exploratory studies and which translated into clinical benefits in relapsing and refractory B cell malignancies [4,5,6,7]. Since then, treatment of other hematologic malignancies has benefited from CAR therapy directed mainly against lineage-restricted proteins [8,9]. However, this approach is hampered in many other cancers, because, unlike B cell tumors, it strictly requires cancer-specific antigens [10]. Despite many efforts to ascertain absence of antigen expression on healthy tissues, the in vivo behavior of infused cells cannot always be fully predicted, potentially leading to patient death [11]. However, it should be noted that epitope refining or careful dosing of the injection product can turn a dangerous CAR into an efficient drug [12]. In addition to problems associated with target recognition, many other parameters may hinder the success of CAR therapy and include product manufacturing, trafficking, infiltration, activation, and persistence. Innovative attempts to control these factors were undertaken by molecular modification of the signaling tail or the combination with associated receptors and have been discussed elsewhere [10,13,14]. An alternative option can be found in exogenous intervention using chemical compounds and approved drugs. Indeed, pharmacologic interventions together with CAR therapy were the center of many investigations during the last decade. In this review, we will discuss the following five pharmacologic approaches involving combinations of CAR T cells with anti-cancer drugs, adverse effect-neutralizing drugs, and drugs used in synthetic system biology to improve CAR clinical outcome (Figure 1): (i) Combinatorial anti-cancer approaches. Multiple pharmacologic modalities have been combined with CAR T cells. Among them are compounds that sensitize cancer cells to apoptosis, Tyrosine Kinase Inhibitors (TKIs), and histone deacetylase inhibitors. (ii) Mitigating adverse effects. Cytokine Release Syndrome (CRS) and neuroinflammation are the main adverse events in CAR therapy. Strategies designed to counteract these effects encompass antagonists or neutralizing monoclonal antibodies (mAbs) directed against cytokines and their receptors (e.g., anti-IL-6R tocilizumab), antagonists of IL-1 receptor (IL-1R), or agents that inhibit macrophage-derived products (nitric oxides). (iii) CAR T cell elimination. Vectors expressing CAR constructs were also designed to harbor tracking (e.g., truncated CD34 or nerve growth factor receptor) and removal systems (e.g., CD20 mimotopes fused to the CD8 stalk for rituximab-mediated antibody-dependent cellular cytotoxicity). An important approach initially developed with other adoptive cell transfer modalities is the use of suicide genes that can induce CAR T cell apoptosis. (iv) Reversible control of CAR. Synthetic biology systems have been exploited to engineer CARs with responsive elements to exogenously control CAR T cell function (e.g., drug-induced dimerization of split CAR format, drug-induced CAR targeting to proteasomal degradation). These approaches are reversible; thus, they are preferred over suicide genes discussed above. (v) Modulating CAR specificity. These modalities mainly express a CAR where the extracellular module is designed to bind an additional exogenously provided soluble recognition module.

## 2. Combinatorial Anti-Cancer Approaches

Transformed cells divert normal cellular growth pathways and evade apoptosis to sustain their survival, which is one of the hallmarks of cancer [15]. Mechanistically, apoptosis inhibitors are commonly overexpressed in tumor cells and involved in drug resistance. These were already early on identified as attractive targets for therapy [16]. Therefore, it was foreseen that the combination of apoptotic drugs with CAR T cells would lead to a synergistic therapeutic effect by inhibiting anti-apoptotic pathways while triggering extrinsic killing pathways. Inhibitors of B cell lymphoma 2 (Bcl-2) family members such as ABT-737 or the orally bioavailable derivative ABT-263/Navitoclax were shown to restore functional intrinsic apoptosis in B cell tumor cell lines induced by CAR therapy, thereby enhancing cytotoxicity [17].

Interestingly, the pre-sensitization of tumor cells with ABT-737 before CAR T cell treatment increased the killing while sparing effectors. The sequential combination seems to be a promising alternative since Bcl-2, Bcl extra-Large (Bcl-xL), and Bcl-2-like protein 2 (Bcl-w) proteins regulate clonal expansion and survival of lymphocytes, and it could impede the function of co-administered effectors. Indeed, Navitoclax induced peripheral thrombocytopenia and T cell lymphopenia in relapsed and refractory lymphoma patients, which was attributed to high-affinity inhibition of Bcl-2 proteins [18]. On a similar basis, suberoylanilide hydroxamic acid (SAHA), a histone deacetylase inhibitor, and Celecoxib, a nonsteroidal anti-inflammatory drug, were used to reverse the development of resistance of non-Hodgkin lymphoma (NHL) cells brought on by the repetitive exposure to CD19-targeting CAR T cells [19]. Furthermore, CD19-resistant NHL cells showing no aberrant loss of CD19 antigen could be sensitized to subsequent CAR therapy and tumor necrosis factor-related apoptosis-inducing ligand (TRAIL) apoptotic pathway upon SAHA and Celecoxib treatment.

In an exploratory study, Dufva et al. screened 526 compounds that spanned several functional classes such as conventional chemotherapy agents, kinase inhibitors, and apoptotic modulators in order to identify inhibitors and enhancers of CD19-targeting CAR T cytotoxicity against B-acute lymphoid leukemia (B ALL) [20]. They identified birinapant, AT-406, and LCL-161, three compounds belonging to the family of second mitochondria-derived activator of caspase (SMAC) mimetics or inhibitors of apoptotic antagonists to be the most potent molecules combined with CAR T cells. Other molecules such as the protein kinase C (PKC) modulator bryostatin-1, E3 ubiquitin–protein ligase MDM2 inhibitors (idasanutlin and nutlin-3), and topoisomerase 2 inhibitors (etoposide and teniposide) also enhanced cytotoxicity. Furthermore, a CRISPR–Cas9 screen highlighted that signaling through the death receptors TRAIL, tumor necrosis factor (TNF), and especially Fas-associated death domain-containing protein (FADD) is a key aspect of CAR T cell cytotoxicity. In addition, sensitizers with regulatory effects on apoptotic genes can rescue antigen-independent tumor resistance, such as resistance caused by the loss of death receptor signaling or by immunosuppressive signals emitted by the tumor microenvironment (TME) [14,21]. The effect of the microenvironment is more complex in solid tumors, which might therefore greatly benefit from pharmacological sensitizing. Although further investigation of in vivo efficacy of small molecules/CAR combinations with careful dissection of T cell interactions and persistence is needed to define optimal treatment schedules, targeting the apoptotic machinery to sensitize tumor cells to CAR T cells is appealing. To our knowledge, only CD19-targeting CAR T cells have been investigated thus far, and further studies will reveal whether a wider applicability is possible.

Since infused CAR T cells may display an “exhausted” progressive loss of function partly due to inhibitory receptor expression (e.g., PD-1 and Tim-3) [22,23], immune checkpoint blockade was also tested in combination with CAR therapy [24,25,26]. For instance, the specific blocking of PD-1 immunosuppression enhanced epidermal growth factor receptor 2 (HER2)-targeting CAR T cell function while diminishing the occurrence of myeloid-derived suppressor cells within the tumor [24]. Therefore, the use of monoclonal antibodies can significantly restore CAR T cells functionality. Recent reviews have already covered the subject, which will not be further detailed here [27,28,29]. In addition, cytokines have also been scrutinized along CAR T cell expansion and appeared to shape the differentiation state and thus enhance the effector function. For instance, IL-2 has been shown to promote effector-memory and terminally differentiated effector T cell phenotypes in comparison to IL-15 that more often polarizes CAR T cells to stem cell memory phenotype with a higher proportion of cells (both CD4^+^ and CD8^+^) [30,31,32]. The metabolic fitness of CAR T cells was also improved, which resulted in superior in vivo antitumor activity. The use of other cytokines, such as IL-12 or IL-18, has been investigated in the context of fourth generation “armored” CAR T cells, featuring more potent activity against refractory solid cancers with strongly immunosuppressive TME [33,34]. Therefore, cytokine interventions may enhance CAR therapy but will likely have to be carefully dosed to avoid adverse effects (AEs) (see next section).

Another axis is the targeting of hijacked and constitutively active signaling pathways that sustain tumor proliferation, angiogenesis, and metastasis. To this end, selective small molecule TKIs and serine/threonine kinases inhibitors were developed such as trametinib (mitogen-activated protein kinase kinase (MEK) inhibitors) or vemurafenib and dabrafenib (anti-BRAF) [35]. The rational of using vemurafenib has been highlighted in tumor-infiltrating lymphocytes and TCR-modified T cells, where it promoted antitumor responses in BRAF_V600E_ mutated cancers [36,37]. Therefore, this modality was also investigated in melanoma in combination with a CAR [38]. Unlike previous studies, disialoganglioside (GD2)-targeting CAR T cells showed reduced functionality when combined with vemurafenib, whereas trametinib and dabrafenib showed no impairment of the CAR T cells. Conversely, PI3K/Akt/mTOR pathway inhibition during CAR T cells expansion improved antitumor cell function due to enhanced T-helper 1 cytokine polarization [39]. Otherwise, ibrutinib, a Bruton’s tyrosine kinase (BTK) inhibitor, improved engraftment of CAR T cells, tumor clearance, and survival in human or xenografts models of BTK-resistant acute and chronic lymphocytic leukemia [40,41,42,43]. Among others, sunitinib [44], crenolanib [45], and midostaurin [46] TKIs were also combined with CAR therapy. Although midostaurin was found to inhibit CAR T cells, crenolanib and sunitinib led to superior antitumor responses. These studies support the possibility of using TKIs to promote the antitumor effect before or simultaneously with CAR-antigen challenge. As a result of potential interference with biological processes that are also important in T cells, the choice of TKI should be carefully assessed to avoid dampening therapy efficacy to suboptimal levels. However, this suppressive side effect has also been used as an advantage to successfully pause CAR function (see the section reversible spatio-temporal control of CAR). Additional data are required to understand the scope of kinase inhibition to fully define the suitability of TKI and CAR combination therapy.

## 3. Mitigating the Adverse Effects

Although the first-in-human trials involving CD19-targeting CAR T cells have shown remarkable anti-cancer responses, associated toxicities due to the infusion of cells have somewhat tempered enthusiasm [6,7]. Indeed, life-threatening AEs have been observed in addition to the expected long-lasting B cell aplasia. These AEs were found to be associated to tumor burden [47,48] and promoted by conditioning chemotherapy [49,50]. The two most common AEs are CRS, which are manifested by fever, hypotension, and respiratory distress associated with high level of serum cytokines; and neurotoxicity, which is mainly characterized by cognitive disorders, encephalopathy, and probable seizure [47,48,51,52,53]. The high occurrence of severe AEs in clinical CAR T cell studies may hamper wide applicability of this form of immunotherapy if appropriate safety measures are not in place.

The development of CRS often occurs within a few days after CAR T cell injection and has been linked to cell expansion in vivo. Effector cells massively produce cytokines (including GM-CSF, IL-1, -2, -6, IFN-γ, MCP-1, TNF-α) upon interactions within the complex immune environment of tumor patients from which monocytes and macrophages were identified as a major IL-6 producer [54]. Early management of CRS mainly relied on the use of anti-IL-6 strategies, such as the FDA-approved monoclonal antibody tocilizumab, which targets the IL-6 receptor [50,55,56,57,58], and to a lesser extent the anti-IL6 antibody siltuximab [55,59]. Although efficient at mitigating CRS-related AEs, tocilizumab has failed at preventing delayed neurotoxicity that may occur after the onset of CRS symptoms [47,50,60]. At present, there is no approved and widely effective therapy to improve neurotoxicity, and high doses of systemic corticosteroids are often given to weaken the overall immune response, possibly interfering with CAR T cells’ persistence and functions [55]. For instance, dexamethasone is commonly used as first-line treatment due to its efficient penetration of the central nervous system [61]. Patient serum analysis also identified high levels of GM-CSF [6], which can be countered by the GM-CSF neutralizing monoclonal antibody lenzilumab. Pharmacological blockade of GM-CSF in combination with CD19-targeting CAR T cells in an ALL-patient-derived xenograft model showed prevention of CRS and neuroinflammation. Interestingly, CAR T cells knocked-out (KO) for GM-CSF enhanced their antitumor functions, suggesting a parallel approach to control GM-CSF production [62,63]. Others have studied the neutralization of IFN-γ and IL-6 using specific mAbs and showed an indirect reduction of weight loss-associated toxicity in mice [64]. The targeting of TNF-α has also been investigated with etanercept, a soluble TNF-α receptor, and infliximab, a neutralizing mAb [50].

CAR T-associated toxicities have also been addressed using small molecules. A noteworthy example is anakinra, an IL-1 receptor antagonist, used concurrently with nitric oxide inhibitors (L-NIL or 1400 W) or tocilizumab to inhibit macrophage-derived products that contribute to CRS. These combinations abrogated CRS- and neurotoxicity-related mortality while at the same time extending leukemia-free survival [65,66]. Anakinra is currently being tested in different phase-2 clinical trials (NCT04150913, NCT04205838, NCT04359784, NCT04432506) for the prevention of CRS and neurotoxicity in multiple B cell malignancies treated with either CD19- or BCMA-targeting CAR T cells, as well as in prostate cancer treated with prostate-specific membrane antigen (PSMA)-targeting CAR T cells. Finally, treatment with the tyrosine hydroxylase inhibitor metyrosine, which blocks the synthesis of catecholamines, protected mice from lethal complications of CRS induced by CD19-targeting CAR T cell infusion [67]. This study also found that catecholamines promote inflammation by a self-amplifying feed forward loop of cytokine release in myeloid and T cells.

Taken together, diverse pharmacologic approaches have been investigated to mitigate the AEs based on insights gained from clinical CAR T cell studies. The toxicities are not fully understood, but they converge toward the identification of common traits requiring fine monitoring of patients likely to undergo fatal outcomes if not carefully managed. The above-mentioned drugs are potent (e.g., high-doses of immunosuppressive agents) and therefore themselves not devoid of iatrogenic AEs, which are capable of inducing irreversible sequelae if used over a long period. Furthermore, such drugs were not primarily designed to eradicate the cell therapy product and rather indirectly correct the symptoms of AEs. As discussed in the next section, the prophylactic incorporation of safety systems embedded in the CAR technology allows for better control of treatment.

## 4. CAR T Cell Elimination

A drastic method to overcome high toxicities associated with cell-based therapy is to specifically remove the therapeutic cells. Prominent early strategies relied on suicide genes, which control T cell fate toward the termination of DNA replication and subsequent cell death. To this end, expression of herpes-simplex virus thymidine kinase (HSV-TK) or human mutated thymidylate kinase (hMTK) have been used to induce cell depletion. HSV-TK sensitizes cells to ganciclovir, whereas hMTK renders cells sensitive to the pro-drug 3′-azido-3′-deoxythymidine (AZT) [68,69,70]. The usefulness of HSV-TK has been validated in clinical trials showing efficient cell depletion [71]. However, since HSV-TK contains virus-derived immunogenic sequences and was shown to evoke xenoresponses, MTK, being human, is currently preferred although appearing less efficient [72,73].

More recently, pro-apoptotic molecules such as Fas, FADD, the death effector domain (DED) of FADD, or caspase 9 were engineered for inducible activation upon the addition of chemical inducers of dimerization (CID) [74,75,76,77]. It is noteworthy that Straathof et al. used a modified human caspase 9 fused to FK506 binding protein (FKBP) to allow conditional dimerization upon the addition of a non-toxic FK506 analog (AP20187/AP1903) [77]. These initial studies showed that up to 99% of transgenic T cells were depleted following pharmacological treatment [74,77]. As a consequence, the treatment was able to selectively reverse Graft-versus-Host disease (GVHD) mediated by haploidentical T cell transplants in leukemia and lymphoma patients [76,78]. Several CAR studies have harnessed inducible caspase 9 (iCasp9) safety switches [79,80,81]. For instance, in a humanized mouse model of B lymphoma, a direct application of the above-mentioned system together with CD19-targeting CAR expression allowed for dose-dependent containment of redirected T cells with normal B cell reconstitution after the addition of AP1903 (also named rimiducid) [79]. The EMA- and FDA-approved drug rapamycin has also been used to induce the dimerization of caspase 9 by bringing together the FKB-rapamycin binding domain (FRB) fragment of mammalian target of rapamycin (mTOR) and FKBP, thereby leading to selective in vivo ablation of CD19-targeting CAR T cells [82]. Several pre-clinical and clinical studies have explored the iCasp9 system to control CAR T cells, including CD19- (NCT03016377, NCT03696784) GD2- (NCT04196413, NCT01953900, NCT01822652) and Mesothelin-targeting CAR T cells (NCT02414269).

The use of surrogate markers such as truncated CD20 or CD34 were used to control and detect, respectively, the CAR T cell population [83,84,85,86]. Here, the exogenous addition of an authorized therapeutic monoclonal antibody, rituximab, enabled cell tracking and in vivo depletion of CAR T cells by antibody-dependent cellular cytotoxicity (ADCC). An alternative strategy is an all-in-one construct, consisting of the fusion of circular CD20 mimotopes with CD34 epitopes associated with the CD8 stalk of an anti-GD2 CAR, which allows for epitope-based targeting with rituximab on the same molecule [87]. Another approach selected a truncated form of human epidermal growth factor receptor (EGFR) for specific cell deletion upon cituximab (anti-EGFR mAb) treatment [88]. This strategy has been applied along CD19 CAR expression in hematopoietic stem cells (HSCs) [89].

Taken together, in case of AEs, these transgenic approaches encoding suicide switches or mAb-selectable markers are intended to eradicate effector cells by irreversibly removing the cell therapy product. However, despite encouraging results, none of these technologies were entirely efficient in depleting cells, possibly causing long-lasting background toxicity [72]. Although iCasp9 and CD20 transgenes have displayed immediate cell-death induction after in vitro pharmacologic treatment [72], inhibition of the infused product might not occur quickly enough to prevent the onset of AEs. Therefore, other approaches that fine-tune the spatial and temporal presence of the CAR itself in T cells have been the center of many investigations in the past 5 years to reduce the toxicities and redirect cellular specific functions.

## 5. Reversible Spatio-Temporal Control of CAR

The main signaling component found in the intracellular tail of CARs is the CD3ζ subunit of the TCR signaling complex [1,90]. CARs signal through CD3ζ involving multiple phosphorylations of SRC family kinases [91,92,93], which, when activated, trigger the activation of a series of pivotal signaling proteins, ultimately leading to the induction of critical transcription factors. Similar to TCR activation, CAR stimulation affects T cell fate by inducing important changes that eventually lead to cytotoxic activity, differentiation, or expansion. Therefore, CARs might be sensitive to kinase inhibitors targeting the TCR signaling cascade and interfering with this signaling pathway might temper CAR-dependent immune cell activation. To this end, TKIs have been evaluated in CAR T cells (Figure 2i). Dasatinib is an EMA- and FDA-approved BCR–ABL TKI for the treatment of chronic myeloid leukemia and Philadelphia-positive ALL [94,95]. Through an off-target effect on SRC kinases, this TKI additionally suppresses T cell activation, which has been exploited to tune CAR activity [20,96,97,98]. In mouse xenograft models of ALL and lymphoma, dasatinib potently ablated the signaling of CD19-targeting CAR T cells, leading to the suppression of cytotoxicity, cytokine secretion, and proliferation [99,100]. Importantly, the small molecule also protects against fatal CRS in a mouse model of CRS [99]. The onset of action is rapid, but it is also quickly reversible upon the discontinuation of dasatinib, and this conservation of therapeutic potential of CAR T cells is an important advantage of this strategy. Although pharmacologic schemes and doses remain to be settled, the extensive use of dasatinib in onco-hematology, and the management of its related AEs render the molecule attractive for clinicians. Other SRC-inhibiting TKIs such as ponatinib, and saracatinib, FLT3-inhibiting TKIs such as midostaurin, but also MAPK pathway inhibitors refametinib and trametinib, and the calcineurin inhibitor tacrolimus, all known in the clinic, strongly suppressed CAR cytotoxicity through signaling inhibition [20,46].

Another reversible strategy to control CAR T cell function is to regulate the expression of the transgene through inducible vectors responsive to drugs (Figure 2ii). Tetracycline (Tet)-ON/Tet-OFF systems have been developed in which CAR T cells can be activated in the presence or absence of a tetracycline analog [101,102,103]. Although the system featured high inducibility upon doxycycline treatment, vector leakiness and important time-to-effect issues have raised safety concerns, particularly when a rapid and absolute shutdown is needed. Furthermore, this system involves potent drugs and proteins of bacterial and viral origin, which could impede vector efficiency due to immunogenicity.

Alternatively, downstream control of CAR structure using small CIDs has been investigated extensively in order to append signaling domains and redirect specialized functions [104,105,106,107,108] (Figure 2iii–vi). One prominent early study harnessed the conditional rapalog-induced dimerization of FKBP/FRB partner proteins for intracellular assembly of a split CAR (Figure 2iii). Wu et al. engineered a tunable ON-switch CAR where the scFv was fused to FKBP, allowing for heterodimerization with membrane-bound signaling domains that were fused to FRB. This split CAR assembled in a strictly rapalog-dependent manner [104], and upon antigen challenge, the CAR T cells proliferated and secreted cytokines only when their receptors were chemically dimerized, which allowed the timing, location, and dosage of T cell activity. This innovation was superior compared to existing control solutions, because the CAR T cell product was not destroyed but rather tuned down. Although the dose and nature of the chemical used in their first report would be difficult to deploy in a clinical setting, the authors described an important proof-of-concept that demonstrated the power of recycling over wasting. They extended their approach by switching the human FKBP/FRB partners to the structurally unrelated *Arabidopsis* gibberellin-induced dimerization domains (GID1/GAI) [109], strengthening the use of chemical orthogonal tools to control CAR activity. Following this study, the extracellular heterodimerization of soluble scFv with membrane-anchored costimulatory domains was also demonstrated using sub-immunosuppressive dose of rapamycin and FKBP/FRB domains [105]. Using the lipid-permeable tacrolimus analog rimiducid, others have investigated the inducible dimerization of MyD88/CD40 to activate downstream Toll-like receptor (TLR) and CD40 signaling [108] (Figure 2iv). Since MyD88 is an essential component of TLR signaling, its redirection was useful to promote survival and proliferation of CAR T cells. These mechanistic designs sought to separate T cell signal 1 (antigen recognition) and 2 (co-stimulation) as a means to safely control signaling potency and proliferation.

The small molecule-based dimerization strategies presented above are powerful tools to artificially regulate interactions between CAR domains, but most of them have important drawbacks that limit their use in humans. Gibberellic acid is plant-derived and therefore likely immunogenic, and rapamycin is toxic and immunosuppressive. Rapamycin-derived rapalogs are less toxic but often show non-favorable pharmacokinetic profiles [104,106,110]. For instance, AP21967 has a half-life of less than 4 h in mouse serum [104]. Therefore, selecting more suitable small molecule and partner candidates with a high degree of orthogonality could overcome these limitations. A recent study addressed this issue by developing antibody-based CIDs (AbCID) that specifically recognize chemical epitopes of CID–protein complexes [106] (Figure 2vi). For example, the Bcl-xL inhibitor ABT-787, which has a favorable serum half-life (14–18 h) [111] has been used in combination with an Ab (Fab AZ1) that specifically recognizes ABT-737-bound Bcl-xL. Then, the authors used a CAR construct in which the scFv portion was replaced by Bcl-xL, which upon ABT-737 addition dimerized with a soluble bispecific antibody made of Fab AZ1 fused to a CD19-targeting scFv. CAR T-transduced Jurkat cells bearing a nuclear factor of activated T cells (NFAT) reporter displayed ABT-737-dependent activation upon antigen challenge with an EC_50_ that was approximately 330-fold lower than the cytotoxic concentration of ABT-737. Although further functional characterization is needed to broaden the applicability to CAR therapy, the study highlights a robust technique to identify novel orthogonal partners. Conversely, Bcl-xL protein has been studied as a form of self-assembling chemically disruptable heterodimers (CDH) able to stop CAR function [112] (Figure 2vii). Bcl-xL was used as a starting point of CDH computational design, since the protein is globular and therefore unlikely to impede T cell synapse configuration, and small molecule partners with a long half-life (> 10 h) are clinically available [113]. The in-silico design sought to incorporate Bcl-xL in the signaling tail of the CAR, while the partner BH3 domain of BIM was placed in the recognition domain. Since BIM is a non-globular, intrinsically disordered protein in an unbound state, the protein was replaced by transplanting the BH3 motif into a human globular protein. The highest affinity Bcl-xL partner retained was the human apolipoprotein E4 (named LD3). Then, a split CAR format was engineered to express an anti-PSMA scFv linked to cytoplasmic CD28-LD3, whereas the CD3ζ chain was separately linked to Bcl-xL. Timed administration of small molecules dynamically and reversely inactivated CAR T cells in an in vivo model of prostate carcinoma allowing for fine-tuning of CAR activity.

Finally, CIDs have been scrutinized to induce conformational changes that orthogonally enable the assembly of high affinity partners (Figure 2v). For instance, a human retinol binding protein 4 (hRBP4) binder was engineered to display around 500-fold higher affinity than hRBP4 upon treatment with a retinol analog [107]. An ON-switch split CAR format was explored in which the soluble CD19-targeting scFv was assembled with a membrane-bound CD28-containing endodomain, successfully permitting the regulation of the activity of primary human CAR T cells in vitro. Several of the synthetic compounds required for the regulation of this CAR are already commercially available as oral drugs, allowing for the rapid implementation of such CARs in clinical settings. As predicted by Wu and colleagues, the replacement of rapalog with non-toxic drugs in a tunable system would be a major advance for future CAR therapies [104].

Unlike the elimination of CAR T cells, another arm to spatiotemporally control cell function is to use rapid proteolytic systems that perturb CAR protein regulation [114] (Figure 2viii–x). Multiple degradation systems have been used as research tools but, similar to CIDs, most of them are not clinically suitable due to (i) their non-human origin and (ii) the toxicity of their pharmacologic partners [115,116,117,118,119]. However, some have recently been considered in CAR technology and highlighted interesting features. It is noteworthy that cleavable degradation moieties (degrons) were incorporated in conventional CAR architecture [120,121,122,123]. Juillerat et al. designed a CD22-targeting CAR bearing a degradation moiety composed of a hepatitis C virus (HCV) NS3 protease target site, the NS3 protease, and the degron in C-terminus [120] (Figure 2viii). Constitutive expression of the CAR induces proteolysis of the degron thus functional state unless Asunaprevir (ASN), an antiviral protease inhibitor, was supplemented (OFF-state). Using the FKBP/FRB pair, GD2-targeting CAR T cells were engineered to harbor a ligand-induced degradation (LID) domain in the form of a cryptic degron [117,122] (Figure 2ix). The specific addition of Shield-1, a synthetic binder of F36V mutant of FKBP12 possessing around 1 000-fold higher affinity than the endogenous FKBP12 [124], displaced and exposed the cryptic degron, which resulted in proteasomal degradation of the CAR and loss of surface expression. In vivo, Shield-1 treatment was able to temporarily reduce CAR T cell activity in tumor-bearing mice, but it required an external injecting device to counterbalance the short small molecule half-life. Using a proteolysis-targeting chimera (PROTAC) against the bromodomain (BD) of brd4 [125], an engineered CD19-targeting CAR was efficiently targeted for E3 ubiquitin ligase-mediated proteasomal degradation, resulting in suppression of CAR T cell cytotoxicity [121]. The removal of PROTAC compounds (ARV-771 and ARV-825) stopped the recruitment of BD-containing CARs to E3 ligases, which reversed the CAR repression. Of note, a recent study has highlighted the importance of E3 ubiquitin ligase recognition sites on a CD19-targeting CAR as a means to sustain or modulate receptor recycling [126]. This approach might find interest in balancing CAR degradation dynamics and could be used as an alternative to the above-mentioned systems to regulate potency and persistence. Another approach further harnessed E3 ubiquitin ligase (CRL4^CRBN^) in combination with Cys^2^-His^2^ (C2H2) zinc finger degron motifs to allow drug-dependent interaction upon the addition of FDA- and EMA-approved thalidomide analogs [123,127]. An OFF-switch CAR was designed as a single chain molecule carrying the zinc finger degron, which was targeted for proteasomal degradation when lenalidomide was added (Figure 2x). Conversely, ON-switch CAR was also considered with two chains: one was composed of signaling domains fused to E3 ligase, whereas the other bore the scFv linked via a transmembrane domain to the degron. Here, lenalidomide induced split CAR heterodimerization, switching it to a functional ON-state. The authors showed that the same modules could be exploited to either induce or repress the function of a CAR. Although attractive in using subtherapeutic doses of thalidomide analogs, this method showed some limitations, the main one being the immunogenicity of their enhanced degron motifs.

Altogether, these proteolytic approaches involve advanced synthetic biology designs that feature rapid spatio-temporal control of CARs in comparison to gene-regulated systems. Most of them sought to manipulate approved drugs to accelerate their clinical use. However, small pharmacologic molecules may also display drawbacks such as a paucity of tissue specificity responsible for an extended diffusion in the body and, in some cases, of cell penetration to reach the target. Beyond the use of small molecules, it is worth mentioning recent efforts to provide non-invasive control over the time, location, and dosage of CARs using either reversible optogenetic or rational designs of dual antigen sensing, such as inhibitory CARs or synNotch positive-feedback circuits sensing antigen threshold [128,129,130,131].

## 6. Modulating CAR Specificity

Since CAR specificity relies on the scFv-antigen recognition, different research groups have shown that it is possible to exchange this part by the exogenous addition of a scFv module. This strategy intends to not only control the timing and dosing of CAR function but also to expand the application range by considering the therapeutic cells as a modular recognition platform adaptable to different tumors, which is an important requirement for the universal implementation of the method.

Switching and/or adding separate antigen recognition has been studied through the engineering of soluble scFvs and mAbs. Initial strategies used antibodies that were non-specifically or enzymatically labeled with haptens (incomplete antigens potentially immunogenic with a carrier protein, such as Ab) to redirect anti-haptens CAR T cell activity [132,133,134,135]. For instance, rituximab, cetuximab, or trastuzumab (an anti-human HER2 mAb) modified to contain fluorescein isothiocyanate (FITC) efficiently redirected anti-FITC CAR T cells, which was further attenuated upon non-specific anti-FITC IgG addition [132]. Potent and switchable antitumor activities were also achieved using CD19-targeting scFv or folate conjugated with FITC, underscoring the versatility of this strategy to target various tumor-associated antigens (TAAs) [132,134,135,136]. However, the use of haptens may induce the production of neutralizing anti-hapten Abs in treated patients or, at a molecular level, sterically obstruct T cell synapse formation, both impeding efficient CAR therapy [135,137]. In line with this approach, others have exploited epitope tags to redirect anti-tag CAR T cells using engineered soluble modules. For example, using anti-CD33 and anti-CD123 scFvs fused to an epitope derived from the human nuclear autoantigen La/SS-B, anti-epitope CAR T cells were able to kill AML blasts in vivo [138,139]. The technology, named UniCAR T for the switchable universal CAR T platform, sought to target antigens simultaneously or subsequently to dampen the risk of selecting tumor variants. An additional technology was proposed by the same group in which the soluble modules consisted of anti-TAAs and anti-tag bispecific antibodies combined with an anti-tag CAR [140]. This approach was named RevCAR (for reversible CAR), where combinatorial Boolean logic (e.g., “OR” and “AND”) could be achieved with rapid turn on/off kinetics. The universal CAR setting was tested against multiple antigens such as PSMA, prostate stem cell antigen (PSCA), FLT3, and EGFR, showing good efficiency [141,142,143]. However, although elegant and innovative, this technology might face the same challenges as the bispecific antibody-based therapy such as the short half-life of the injected product [144,145]. Antibodies modified with peptide neo-epitopes (PNE) also enabled retargeting of anti-PNE CAR T cells toward TAAs. This strategy was particularly adopted to address antigen loss in B ALL variants [137,146]. Another notable innovative approach took advantage of soluble scFv fused to leucine zipper motifs: the split, universal, and programmable (SUPRA)-CARs [147]. The system provides versatile ON/OFF-switch possibilities as well as combinatorial Boolean logic responses by tuning zipper motif affinity and scFv specificity. A substantial number of studies exploiting the modular potential of soluble antibodies with universal CARs is emerging [148,149,150] and will therefore not be further detailed here, since a comprehensive review has recently been published [151]. Of note, the above-mentioned systems solely rely on an antibody fragment that, in a near-cell environment, binds the extracellular CAR domain without additional control. A recent study used a unique nanobody scaffold [152] to modulate CAR recognition and activation with methotrexate (MTX) [153]. This elegant approach used conditional scFvs that specifically bound TAAs in the absence of MTX and display reduced binding in the presence of MTX allowing exogenous control over CAR T cell function using an approved therapeutic molecule. CD33- and EGFR-targeting CAR T cells achieved cytotoxicity to a level comparable to conventional single-chain CARs while being reversibly attenuated by the addition of MTX. This study merged efforts to redirect specificity, attenuate CAR therapy-associated AEs, and could be applied to a broad range of antibody-based therapeutics.

Overall, these adaptable approaches pave the way for tailored targeting, especially for the treatment of minimal residual disease and relapse reminiscent to suspected antigen loss. The modular systems discussed herein aim at offering adapted and cheaper product to the patients. In addition, these methods can also target simultaneously or sequentially several TAAs, which could improve the clinical outcome as recently exemplify here [154].

## 7. Concluding Remarks

CAR T cells have shown unprecedented results in treating cancers otherwise not curable. However, the therapy is still not optimal. This is partially due to the nature of the product, which is a “living drug”, suggesting that its control might vary with its origin (patient fitness) and quality. It is worth mentioning that so far, all CAR studies have at least identified severe grade 3 or 4 clinical AEs and/or patient death. We have in the present review highlighted novel, innovative approaches to deal with the versatility of CAR T cells by combining them with exogenous pharmacologic drugs (see Table 1 for an overview).

The first obvious utility unites the effect of drugs to sensitize tumor cells with CAR therapy in order to further synergize the effects of the two treatments. This arm should reveal its full potential to alleviate tumor resistance mechanisms and counteract the non-permissive TME. However, double treatment also means more and often unexpected side effects, and careful monitoring of the patients will be required during the initial validations. Another important use of drug combination was found in approaches inducing immune cell suicide. Although elegant and able to save patients from undesirable effects, drugs can be costly. However, compared to the generation of patient-compatible transgenic T cells, drug costs will likely only be a minor part of total treatment costs. We predict that more advanced switch technologies will emerge in which the precious therapeutic material will be spared by carefully modulating activity of the CAR rather than by destroying the product. Tuning the therapeutic effect of the CAR will be an important step toward a personalized adaptation of the “living drug”. In line with this, several groups have used FDA and EMA-approved drugs that could readily be transferred to the clinic to control CAR function. This has created a positive synergy between system biologists and immunotherapists with the ambition of identifying innovative systems with specific pharmacologic partners.

In conclusion, the success of CAR therapy relies on better control of the product as well as on treatment personalization. We herein discussed one strategy: the combination of pharmacological drugs with CAR treatment and observed that it was innovative, elegant, and feasible. Therefore, we expect to see future CAR/drugs trials showing efficient clinical outcomes in the near future.

## Figures and Tables

**Figure 1 ijms-22-04320-f001:**
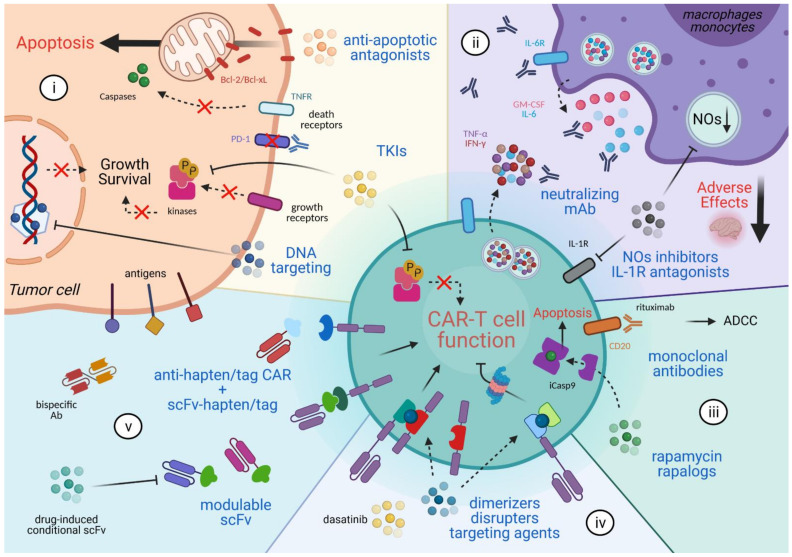
Overview of the pharmacologic interventions exploited with CAR T cells. (i) Combinatorial anti-cancer approaches. (ii) Mitigating adverse effects. (iii) CAR T cell elimination. (iv) Reversible control of CAR. (v) Modulating CAR specificity. See the main text for description. Created with BioRender.com.

**Figure 2 ijms-22-04320-f002:**
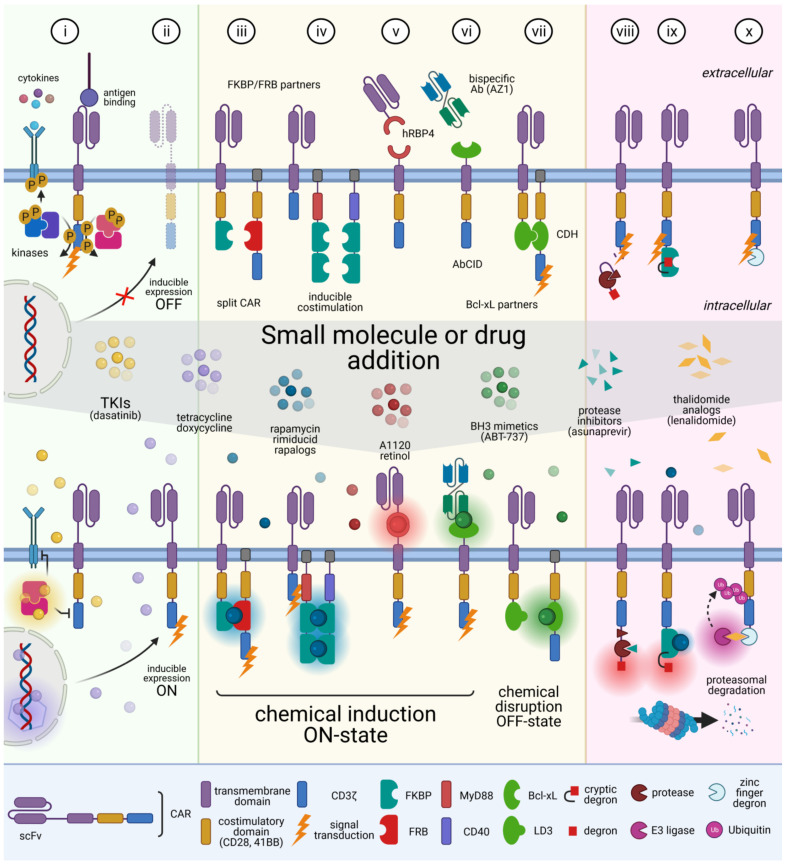
Pharmacologic strategies to reversibly control CAR function. Engineered CAR T cells can be commanded through signaling pathways and transgene expression (left panel), structural CAR component interactions (middle panel) or targeting CAR for proteolysis (right panel). See the main text for detailed descriptions. Created with BioRender.com.

**Table 1 ijms-22-04320-t001:** Selected studies combining CAR T cells and small molecules or drugs.

Year	CAR Target	CAR Design *	Cancer or Model	Small Molecule or Drug	Target	Observations	Ref.
**Combinatorial anti-cancer approaches**							
2013	CD19	scFv-CD28-CD3ζ	B ALL	ABT-737; ABT-263 (navitoclax)	Bcl-2 family members	Restore intrinsic apoptosis in tumor cells; Enhance CAR T cells efficacy	[17]
2018	CD19	N/A	NHL	suberoylanilide hydroxamic acid and LBH589; Celecoxib	Histone deacetylase; cyclo- oxygenase-2	Enhance CAR T cells cytotoxicity	[19]
2020	CD19	scFv-CD28-CD3ζ	B ALL	>500 small molecules	Multiple	primary: birinapant, AT-406, LCL-161 (SMAC mimetics/inhibitor of apoptotic antagonists); secondary: bryostatin-1 (PKC activator), idasanutlin and nutlin-3 (MDM2 inhibitors), etoposide and teniposide (topoisomerase 2 inhibitors); Enhance CAR T cells cytotoxicity	[20]
2013	HER-2	scFv-CD28-CD3ζ	HER-2^+^ PD-1^+^ tumor cells	anti-PD-1 mAb	PD-1	Enhance CAR T cell function; decrease MDSCs frequency	[24]
2017	CD19	scFv-CD28-CD3ζ	B ALL	Akt inhibitor VIII	Akt	Akt signaling inhibition during CAR T cell expansion improve antitumor efficacy	[39]
2013	CD19	scFv-CD28-CD3ζ	CLL	ibrutinib	Bruton’s tyrosine kinase	Improve CAR T cells engraftment, tumor clearance and mice survival	[41]
2020	CAIX	scFv-41BB-CD3ζ	RCC (lung metastasis)	sunitinib	Multiple kinases	Up-regulate CAIX in tumor cells; decrease MDSCs frequency	[44]
2018	FLT3	scFv-CD28/41BB -CD3ζ	AML	crenolanib	FLT3 kinase	Synergize anti-leukemia effect	[45]
**Mitigating the adverse effects**							
2016	CD19	scFv-41BB-CD3ζ	B ALL	etanercept, infliximab	TNF-α	Reduce toxicity	[50]
2019	CD19	scFv-41BB-CD3ζ	B ALL	lenzilumab	GM-CSF	Suppress CRS and neurotoxicity	[62]
2018	CD19	scFv-CD28-CD3ζ	B lymphoma	anti-IL-6 and anti-IFN-γ mAb	IL-6 and IFN-γ	Reduce toxicity	[64]
2018	CD19, CD44v6	scFv-CD28-CD3ζ	B ALL	anakinra, tocilizumab	IL-1 receptor antagonist,IL-6	Suppress CRS and neurotoxicity; Extend leukemia-free survival	[65]
2018	CD19	scFv-CD28-CD3ζ	B ALL	anakinra; L-NIL and 1400W	IL-1 receptor antagonist, iNOs inhibitors	Inhibit macrophage-derived products (NOs, IL-1 and IL-6); Suppress CRS-related mortality	[66]
2018	CD19	scFv-CD28-41BB -CD3ζ	B lymphoma	metyrosine	catecholamines	Protect mice from lethal complications of CRS	[67]
**CAR T cell elimination**							
2017	CD19	iCasp9-2A-tNGFR -2A-scFv-41BB -CD3ζ	B lymphoma	AP1903 (rimiducid)	FKBP/FRB Inducible caspase 9 (iCasp9)	Eliminate CAR T cells in a dose-dependent manner	[79]
2018	CD19	rapaCasp9-2A -RQR8-2A-scFv -41BB-CD3ζ	B ALL and lymphoma	AP20187, rapamycin	FKBP/FRB Inducible caspase 9 (rapaCasp9)	Eliminate CAR T cells in vivo	[82]
2014	GD2	N/A	N/A	rituximab	CD20 epitope fused to CD8 stalk (RQR8, also contain tCD34)	Enable CAR T cells selection, cell tracking (tCD34) and deletion (CD20)	[87]
**Reversible spatio-temporal control of CAR**							
2019	CD19	scFv-CD28/41BB -CD3ζ-2A-EGFRt	B lymphoma	dasatinib	SRC kinases	Reversibly suppress CAR T cell cytotoxicity, cytokine secretion, and proliferation; protect from CRS	[99]
2019	CD19	scFv-CD28/41BB -CD3ζ	B ALL	dasatinib	SRC kinases	Reversibly suppress CAR T cell cytotoxicity, cytokine secretion, and proliferation	[100]
2015	CD19, Meso	scFv-41BB-FKBP + DAP10-41BB -FRB-CD3ζ	CD19^+^ or Meso^+^ K562	rapalog, (gibberellic acid)	FKBP/FRB- (or GID1/GAI) -based CAR dimerization	ON-switch CAR: control the timing, location, and dosage of CAR T cell activity; mitigate toxicity	[104]
2018	CD19	soluble bispecific scFv-Fab(AZ1) + BclxL-41BB -CD3ζ	CD19^+^ K562	ABT-737	Fab(AZ1) specific for Bcl-xL only in the presence of ABT-737	Drug-dependent CAR T cell activation	[106]
2020	CD19	scFv-IgG1Fc -hRBP4 + RS3-IgG1Fc -CD28-CD3ζ	B ALL	A1120	hRBP4 and hRBP4 binders (RS3)	Drug-dependent regulation of CAR T cells activity	[107]
2017	PSCA, GD2, CD123	iMC: MyD88-CD40 -(FKBP)_x2_-2A -ΔCD19 iMC-2A-scFv -CD3ζ scFv -CD28/41BB/OX40-CD3ζ	prostate, melanoma, AML	AP1903 (rimiducid)	FKBP/FRB -based dimerization of MyD88/CD40 (iMC)	Enhance CAR T cell proliferation and antitumor activity	[108]
2020	PSMA	scFv-CD28-LD3+ DAP10-CD28 -BclxL-CD3ζ	prostate	A-1155463, A-1331852 (BH3 mimetic)	LD3/Bcl-xL -based CAR dimerization	STOP-CAR: dynamically and reversibly inactivate CAR T cells	[112]
2019	CD22	scFv-41BB-CD3ζ -NS3cleaving_site -NS3protease -degron	B lymphoma	Asunaprevir	HCV NS3 protease	Switch-OFF CAR (SWIFF-CAR): constitutive CAR degron proteolysis; Asunaprevir-dependent CAR degradation	[120]
2020	CD19	scFv-CD28-BD2 -CD3ζ	B ALL	ARV-771 or ARV-825 (retinol)	bromodomain (BD of brd4)	Induce drug-dependent CAR degradation; Reversibly suppress CAR T cells	[121]
2020	GD2	scFv -41BB/KIR2DS2 -CD3ζ-[FKBP -degron]_LIDdomain_	FAP^+^ Mesothelioma	Shield-1	LID domain -based CAR degradation	Induce drug-dependent CAR degradation; temporarily reduce CAR T cells activity	[122]
2021	CD19	OFF-switch: scFc-41BB-CD3ζ -C2H2degron ON-switch: CD8-CD28-CRBN-CD3ζ + scFv-CD28 -C2H2degron	B ALL and lymphoma	thalidomide analogs	C2H2, CRBN	OFF-switch: thalidomide analog-induced CAR proteasomal degradation, limit inflammatory cytokine production while retaining antitumor efficacy ON-switch: thalidomide analog-induced split CAR dimerization, drug-dependent antitumor activity	[123]
**Modulating CAR specificity**							
2016	FITC	scFv-41BB-CD3ζ	B ALL and lymphoma	FITC-modified anti-CD19 and anti-CD22 antibodies	CD19, CD22	Enable CAR-switch combinations; potent and dose-dependent antitumor activity	[135]
2016	5B9 epitope of La/SS-B	scFv-CD28-CD3ζ	AML (others)	5B9-tagged anti-CD33 and anti-CD123 antibodies	CD33, CD123	UniCAR T (Universal): Redirect CAR in a time- and target-dependent manner; potent anti-AML activity	[139]
2018	HER-2, Axl, Meso	zipFv: scFv -EEleucine_zipper zipCAR: RRleucine_zipper -CD28-41BB-CD3ζ	HER-2^+^, Axl^+^, Meso^+^ K562	Soluble zipFv	Membrane -bound zipCAR	SUPRA-CAR: control signaling, fine-tune T cell activation, mitigate toxicity and allow multiple antigens sensing	[147]
2021	CD33, EGFR	scFv-41BB-CD3ζ	AML, GBM	Methotrexate	Conditional scFvs	Drug-induced decrease of CAR T cells affinity and cytotoxicity; reversible	[153]

* Only the recognition, costimulatory, and signaling domains are depicted in order to facilitate the understanding. Additional leader sequence, hinge/stalk, transmembrane domains, tags, and epitopes were removed when irrelevant. A slash (/) means that either one or more of the costimulatory domains is present. N/A: Non-available. AML: acute myeloid leukemia; B ALL: precursor B acute lymphoid leukemia; Bcl-2: B cell lymphoma 2; Bcl-xL: Bcl extra-large; C2H2: Cys^2^-His^2^; CAIX: carbonic anhydrase IX; CLL: chronic lymphoid leukemia; CRBN: cerebelon; CRS: cytokine release syndrome; EGFR: epidermal growth factor receptor; FKBP: FK506 binding protein; FLT3: fms-like tyrosine kinase 3; FRB: FKBP–rapamycin-binding; GBM: glioblastoma multiform; GD2: disialoganglioside; GM-CSF: granulocyte-macrophage colony stimulating factor; HCV; Hepatitis C virus; HER-2: epidermal growth factor receptor 2; hRBP4: human retinol binding protein 4; IFN-γ: interferon gamma; iMC: inducible MyD88/CD40; La/SS-B: human nuclear auto-antigen La/SS-B; LD3: human apolipoprotein E4; LID: ligand-induced dimerization; MDM2: mouse double minute 2 homolog; MDSC: myeloid-derived suppressor cells; Meso: mesothelin; NHL: non-Hodgkin lymphoma; NOs: nitric oxides; PD-1: programmed cell death protein 1; PKC: protein kinase C; PSMA: prostate-specific membrane antigen; PSCA: prostate stem cell antigen; RCC: renal cell carcinoma; scFv: single-chain variable fragment; SMAC: second mitochondria-derived activator of caspase; tCD34: truncated CD34; TNF-α: tumor necrosis factor alpha; tNGFR: truncated nerve growth factor receptor; zip: leucine-zipper.

## Data Availability

Not applicable.

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
