# Peer review of "Pharmacologic Control of CAR T Cells"

_ijms, 2021, doi:10.3390/ijms22094320_

Round 1

Reviewer 1 Report

Dear Authors,

your work provides a detailed and state of the art overview of the field. There were only two things I was missing a bit.

a) The combination with Cytokines as for example described in those papers:

Chmielewski M, Abken H. CAR T Cells Releasing IL-18 Convert to T-Bethigh FoxO1low Effectors that Exhibit Augmented Activity against Advanced Solid Tumors. Cell Rep. 2017 Dec 12;21(11):3205-3219. doi: 10.1016/j.celrep.2017.11.063. PMID: 29241547.

Yeku OO, Purdon TJ, Koneru M, Spriggs D, Brentjens RJ. Armored CAR T cells enhance antitumor efficacy and overcome the tumor microenvironment. Sci Rep. 2017 Sep 5;7(1):10541. doi: 10.1038/s41598-017-10940-8. PMID: 28874817; PMCID: PMC5585170.

and Influencing CAR T cells already before implantation as described for example in this paper:

Alizadeh D, Wong RA, Yang X, Wang D, Pecoraro JR, Kuo CF, Aguilar B, Qi Y, Ann DK, Starr R, Urak R, Wang X, Forman SJ, Brown CE. IL15 Enhances CAR-T Cell Antitumor Activity by Reducing mTORC1 Activity and Preserving Their Stem Cell Memory Phenotype. Cancer Immunol Res. 2019 May;7(5):759-772. doi: 10.1158/2326-6066.CIR-18-0466. Epub 2019 Mar 19. PMID: 30890531; PMCID: PMC6687561.

Kind regards

Author Response

We thank the reviewer for her/his comments, we have modified the manuscript accordingly and the papers were cited (l. 149 and l. 153) in addition to some explanatory text.

Reviewer 2 Report

Please see the attached file that contains my review comments.

Author Response

We are grateful for the positive feedback from reviewer 2 which was very helpful. We have taken in account all her/his comments and believe that the paper is now improved.

  • The Figure 1 caption has been shortened and embedded in the main text at the end of the introduction to give an overview of the topics that will further be discussed. (“The current content of caption 1 should be its own separate paragraph in the text at the position that it is at”). Of course, additional information will be found in the main text. Some spelling errors have also been corrected in the Figure 1.
  • The Reviewer also asked to include the different generations of CAR T cells that have been developed with their various motifs. It is indeed an important aspect of CAR therapy but likely falls outside the scope of the current review, which mainly discusses strategies to pharmacologically regulate CAR activity. Related recent reviews have been cited in the text, accordingly. We have added combinations with CAR/cytokines during CAR expansion as suggested and in the context of armored CAR (see section 2, from l.146) without digging into too many details to avoid falling outside of the scope of this review.
  • Grammatical and sentences modifications are underlined in yellow. To further facilitate the understanding of the section 5 (reversible control of CAR), a second figures (figure 2) has been prepared and embedded in the text with citations. Indeed, we realized that this part was dense and might be challenging to understand without schemes.

Reviewer 3 Report

This is an extensive review of the regulation of CAR-T therapy and the future potential of different pathways. I would recommend publication of the review in its present form.

Author Response

We thank the reviewer for her/his positive feedback.